# Semantic Segmentation with Transfer Learning for Off-Road Autonomous Driving

**DOI:** 10.3390/s19112577

**Published:** 2019-06-06

**Authors:** Suvash Sharma, John E. Ball, Bo Tang, Daniel W. Carruth, Matthew Doude, Muhammad Aminul Islam

**Affiliations:** 1Department of Electrical and Computer Engineering, Mississippi State University, Starkville, MS 39762, USA; ss3795@msstate.edu (S.S.); jeball@ece.msstate.edu (J.E.B.); 2Center for Advanced Vehicular Systems, Mississippi State University, Starkville, MS 39762, USA; dwc2@cavs.msstate.edu (D.W.C.); mdoude@cavs.msstate.edu (M.D.); 3Department of Electrical Engineering and Computer Science, University of Missouri, Columbia, MO 65211, USA; mig5g@missouri.edu

**Keywords:** semantic segmentation, transfer learning, autonomous, off-road driving

## Abstract

Since the state-of-the-art deep learning algorithms demand a large training dataset, which is often unavailable in some domains, the transfer of knowledge from one domain to another has been a trending technique in the computer vision field. However, this method may not be a straight-forward task considering several issues such as original network size or large differences between the source and target domain. In this paper, we perform transfer learning for semantic segmentation of off-road driving environments using a pre-trained segmentation network called DeconvNet. We explore and verify two important aspects regarding transfer learning. First, since the original network size was very large and did not perform well for our application, we proposed a smaller network, which we call the light-weight network. This light-weight network is half the size to the original DeconvNet architecture. We transferred the knowledge from the pre-trained DeconvNet to our light-weight network and fine-tuned it. Second, we used synthetic datasets as the intermediate domain before training with the real-world off-road driving data. Fine-tuning the model trained with the synthetic dataset that simulates the off-road driving environment provides more accurate results for the segmentation of real-world off-road driving environments than transfer learning without using a synthetic dataset does, as long as the synthetic dataset is generated considering real-world variations. We also explore the issue whereby the use of a too simple and/or too random synthetic dataset results in negative transfer. We consider the Freiburg Forest dataset as a real-world off-road driving dataset.

## 1. Introduction

Semantic segmentation, a task based on pixel-level image classification, is a fundamental approach in the field of computer vision for scene understanding. Compared to other techniques such as object detection in which no exact shape of object is known, segmentation exhibits pixel-level classification output providing richer information, including the object’s shape and boundary. Autonomous driving is one of several fields that needs rich information for scene understanding. As the objects of interest, such as roads, trees, and terrains, are continuous rather than discrete structures, detection algorithms often cannot give detailed information, hindering the performance of autonomous vehicles. However, this is not true of semantic segmentation algorithms, as all the objects of interests are detected on a pixel-by-pixel basis. Nonetheless, to use this technique, one needs careful annotations of each object of interest in the images along with a complex prediction network. Despite these challenges, there has been tremendous work and progress in object segmentation in images and videos.

Convolutional Neural Networks (CNNs) such as Alexnet [1], VGGnet [2], and GoogleNet [3] have been used extensively in several seminal works in the field of semantic segmentation. For semantic segmentation, either existing classification networks are adopted as a baseline or completely new architectures are designed from scratch. For the segmentation task that uses an existing network as a baseline, the learned parameters on that network are used as a priori information. Semantic segmentation can also be considered as a classification task in which each pixel is labeled with the class of the corresponding enclosing object. The segmentation algorithm can either be single-step or multi-step. In a single-step segmentation process, only the classification of pixels is carried out, and the output of the segmentation network is considered to be the final result. When the segmentation is a multi-step process, the network output is subjected to a series of post-processing steps such as conditional random fields (CRFs) and ensemble approaches. CRFs provide a way of statistical modeling for the structured prediction. In semantic segmentation, CRFs help to improve the boundary delineation in the segmented outputs. Ensemble approaches help to pool the strengths of several algorithms. The results of these algorithms are fused using some rules to achieve better performance. However, these techniques increase the computational cost, making them inapplicable to our problem of scene segmentation for autonomous driving. Therefore, the application of these post-processing steps depends upon the type of domain. The performance and usefulness of the segmentation algorithms are evaluated on the basis of parameters such as accuracy over a benchmark dataset, algorithm speed, boundary delineation capability, etc.

As segmentation holds its importance in the identification/classification of objects, investigating the abnormalities, etc., it applies to a number of fields, such as agriculture [4,5], medicine [6,7], and remote sensing [8,9,10]. A multi-scale CNN and a series of post-processing techniques are applied in [11] to provide a scene labeling on several datasets. The concept of both segmentation and detection is used in [12,13] to classify the images in a pixel-wise manner. Although there has been a lot of work in semantic segmentation, the major improvement was recorded after [14], which demonstrates the superior results on the Pascal Visual Object Classes (VOC) dataset. It performs the end-to-end training and supervised pre-training for segmentation avoiding any post-processing steps. In terms of architecture, it uses the skip layers method to combine the coarse higher-layer information with fine lower-layer information. The methods described in [15,16] are based on an encoder–decoder arrangement of layers that use the max-pooling indices transferred to the decoder part making the network more memory efficient. In both of these works, the mirrored version of the convolutional part acts as the deconvolutional or decoder part. The concept of dilated convolution to avoid information loss due to the pooling layer was used in [17]. A fully connected CRF is used in [18] to enhance the object representation along the boundary. A CRF is used as a post-processing step that improves the segmentation results produced by the network. An enhanced version of [18] is used in [19] which is based on spatial pyramid pooling and the concepts of dilated convolution presented in [17]. A new technique using a pooling called pyramid pooling is introduced in [20] so as to increase the contextual information along with the dilated convolution technique.

All the works mentioned above are evaluated on several benchmark datasets, and one is said to be better than another based on the performance on those datasets. However, in real-life scenarios, there are several areas in which adequate training data are not available. The deep convolutional neural networks require huge amount of training data so that they can generalize well. Lack of enough training data in the domain of interest is one of the main reasons for using Transfer Learning (TL). In TL, the knowledge from a domain, known as the source domain, is transferred to the domain of interest, known as the target domain. In this technique, the deep neural network is first trained in the domain where enough data are available. After this, the useful features are incorporated into the target domain as a priori information. This technique is effective and beneficial when the source and target domain tasks are comparable. The nature of the convolutional neural network to learn general features through lower layers and specific features through higher layers makes the technique of TL effective [21,22]. In particular, in fields such as medicine and remote sensing where datasets with correct annotations are rarely available, the transfer learning technique is a huge benefit. In [23,24], the transfer learning technique is applied for the segmentation of brain structures in brain images from different imaging protocols. Fine-tuning of fast R-CNN [25] for traffic sign detection and classification for autonomous vehicles is performed in [26].

Apart from finding different applications where transfer learning might be used, there has been a constant research effort in effective transfer of knowledge from one domain to another. As it is never the case that all of the knowledge learned from the source task is useful for the target task, deciding what to transfer and how to transfer it holds an important role for the optimum performance of the TL approach. A TL method which automatically learns what and how to transfer from previous experiences is proposed in [27]. A new way of TL for segmentation is devised in [28], which transfers the learned features from a few strong categories, using pixel-level annotations to predict the classes that do not have any annotations (known as weak categories). For a similar transfer scenario, Hong et al. [29] proposes an encoder–decoder architecture combined with an attention model to semantically segment the weak categories. In [30], an ensemble technique, which is a TL approach that trains multiple models one after the other, is demonstrated when the source and target domains have drastic differences.

In our work, we use the TL approach for semantic segmentation specifically for off-road autonomous driving. We use the semantic segmentation network proposed in [16] as a baseline network. This network is trained with the Pascal VOC datasets [31] for segmentation. This domain has a large difference from the one that we are interested in (the off-road driving scene dataset). On the other hand, the off-road driving scene contains fewer classes compared to the Pascal VOC datasets, consisting of 20 classes. Because of this, we propose decreasing the network size, and performing transfer learning on the smaller network. To bridge the difference between the real-world off-road driving scene and Pascal VOC datasets, we use different synthetic datasets as an intermediate domain which might help in performance boosting for the data-deprived domain. Similarly, to correspond to the lower complexity and the latency required for the off-road autonomous driving domain, a smaller network is proposed. Motivated by previous TL approaches in CNN [22,32] and auto-encoder neural networks for classification [33], we transfer the trained weights from the original network to the corresponding layers in the proposed smaller network. However, while most of the state-of-the-art TL methods perform fine-tuning without making any changes to the original architecture (with the exception of the last layer), to the best of our knowledge, this is the first attempt to perform transfer learning from a bigger network to a smaller network, which is helpful to address the two important requirements of autonomous driving. With several experiments using synthetic and real-world datasets, we verify that the network size trained in the source domain may not transfer the best knowledge to the target domain. However, a smaller chunk of the same architecture might work better based on the complexity embedded in the target domain. On the other hand, this work also explores the effect of using various synthetic datasets as an intermediate domain during TL by assessing the performance of the network on a real-world dataset.

The main contributions of this paper are listed as follows:We propose a new light-weight network for semantic segmentation. Basically, the DeconvNet architecture is downsampled to half the original size which performs better for the off-road autonomous driving domain;We use the TL technique to segment the Freiburg Forest dataset. During this, the light-weight network is initialized with the trained weights from the corresponding layers in the Deconvnet architecture;We study the effect of using various synthetic datasets as an intermediate domain to segment the real-world dataset in detail.

The rest of the paper is organized as follows. We briefly review the background and related work in the semantic segmentation of off-road scenes in Section 2. The details of the proposed methods, including Deconvnet segmentation network and our proposed light-weight network, are explained in Section 3. In Section 4, we describe all the experiments and the corresponding results including, the descriptions of the datasets used. Section 5 provides the brief analysis and discussion about the obtained results. The final section of the paper includes our conclusions and notes on future work.

## 2. Background and Related Work

### 2.1. Background

#### 2.1.1. Convolutional Neural Networks (CNN)

The simple CNN architecture is composed of five important layers: the input layer, convolutional layer, activation layer, pooling layer, and fully connected layer. For the purpose of classification, a series of these layers can be used on the basis of the complexity of the dataset under consideration. The convolutional layer extracts the structural and spatial relationships from the image. According to [34], in order to improve the learning task, this layer leverages three important ideas: sparse interactions, parameter sharing, and equivalent representations. The convolutional layer is followed by a sub-sampling layer called the pooling layer. This layer is supposed to capture the high-level information of feature maps in compressed form. Thus, it helps to make the features invariant to smaller transitions and translations which results in CNNs being capable of focusing on the useful properties and ignoring the less important features in the feature space. Max-pooling is the famous pooling technique which takes the maximum value of pixels within a defined boundary as its output. The pooling layer may either alternate with convolutional layer or reside sparsely in the network, depending upon the nature of the classification task.

Another important operation within a CNN architecture is activation. This layer, called the activation layer, introduces the non-linearity in input–output relationship, making CNN a universal function approximator. The last layer in most classification-based CNN architecture is the fully connected layer. The fully connected layer takes the flattened data as input, and is responsible for mixing the signals from each dimension so as to introduce the generalization. However, in most segmentation tasks, this layer is not suitable as it increases the computational cost. CNNs are trained in the same way as multilayer perceptrons, which are trained using back propagation algorithm. Back propagation is based on minimizing the cost function with respect to the weight and adjusting those weights based on the gradient as follows:(1)$$L=\frac{1}{N}\sum_{i}^{N}p\left(y^{i}|X^{i}\right),$$
where N is the total number of images or training samples per batch, $X^{i}$ represents the *i*th input sample, and $y^{i}$ represents the corresponding label. *p*(.) is the probability of correct classification for corresponding input data. For any layer *l*, $W_{l}^{t}$ is the weight vector at *l*th layer at time instant *t*, and $U_{l}^{t}$ is the required update in weight. If $\alpha_{l}$ is momentum, and μ is the learning rate, learning in the network occurs as follows:
(2a)$$\begin{array}{c} U_{l}^{t+1}=\alpha U_{l}^{t}-\mu \frac{\partial L}{\partial W_{l}} \end{array}$$
(2b)$$\begin{array}{c} W_{l}^{t+1}=W_{l}^{t}+U_{l}^{t+1}. \end{array}$$

#### 2.1.2. Transfer Learning

As specified earlier, TL is a way of utilizing the knowledge of a trained model to solve the problem at hand. In the case of the CNN, the network trained on one domain, called the source domain, might have learned some features that would also be relevant to another domain, called target domain. Therefore, the network with the learned features in the source domain could be a better baseline network to accomplish the task in the target domain. Hence, TL involves the use of an existing trained model, modifying its learned features, called knowledge, into target domain features such that it gives acceptable test performance on the target domain. On the basis of this, several TL techniques are notable. In [35], Pan et al. categorize the TL approaches as inductive transfer learning, transductive transfer learning, and unsupervised transfer learning. However, in deep learning, we can also distinguish them differently as: the fine-tuning approach, the feature extraction approach, multitask learning, and meta learning. In the fine-tuning approach, the nature of the CNN to learn general features through the lower layers and specific features through the higher layers is better utilized. The weights learned by the original trained model in lower layers are frozen as they are related to the general properties of images and have greater similarity with the general features of data in the target domain. Only the few higher layers are modified with the dataset in the target domain. The number of higher layers being trained may vary depending on the data distribution differences between the source and target domains. In the feature extraction approach, only the most important features from the source domain that might better represent the features in the target domain are extracted, and the model is trained with those features mixed with target domain dataset. Multitask learning, on the other hand, trains a model on multiple source tasks so as to increase the generalization capability of the network and is finally fine-tuned with the target domain. Meta learning in TL helps the model to learn about what to learn so that the knowledge will be best fitted for the target domain. In this work, we are dealing with a fine-tuning approach.

### 2.2. Related Work

With the advent of powerful GPU technology, CNN-based deep learning techniques have been receiving much attention. Semantic segmentation is one of the fields benefiting from this change. Equally, the interest in intelligent autonomous vehicles has been growing and there has been a large amount of research over recent years. The segmentation of road scenes holds a major role in the functionality of such systems. There have been many works directed at city road environment segmentation. However, there have only been a few works for off-road driving scene segmentation. Daniel et al. perform the semantic mapping for off-road navigation using custom convolutional networks in [36]. In [37], a deep neural network is applied in order to classify the off-road scene as trail and non-trail parts using image patches. It successively applies the dynamic programming to delineate the light-weight trail from sub-light-weight network output. In [38], the TL approach is used to semantically segment the off-road scene using the network trained with on-road scenery. Our work is different in the sense that [16] is trained with Pascal VOC images and we transfer the knowledge to the target, which has very different data distributions. Furthermore, we change the original network size, proposing a smaller network that transfers the optimum knowledge considering the real-time issues required by the autonomous vehicle.

## 3. Proposed Methods

### 3.1. Segmentation Network Structure

The first part of this work aims at finding the light-weight network structure that suits the target domain. This process is largely dependent upon the complexity of the target domain and upon the extent of the source and target domain differences. While designing the autonomous driving systems, two aspects come into play: the safety and processing speed of the autonomous system software. Safety can be seen from a much wider point of view, which is mostly the function of vehicle hardware design and decisions made by the system software. As a result of the nature of autonomous vehicles, a fast processing speed is required for scene understanding and inferencing which ultimately gives robust control over decision making of the vehicle. We consider this requirement to be very important in this work, thus we aim for the smallest possible network size with the highest possible accuracy. In addition to this, transferring all the weights from large pre-trained networks provided sub-optimal results for our synthetic and real-world dataset as the target domains are simpler than the source domain. Therefore, to use the best size of convolutional network (which achieves a suitable processing speed) as well as having an acceptable accuracy level, we propose a smaller convolutional network, called the light-weight network, taking [16] as a base model. Our proposed network, which better suits our application, is half the size of the original Deconvnet architecture. Figure 1 shows the structure of our light-weight network architecture.

The DeconvNet [16] is learned on top of the VGG-16 network [2] and takes 2D images 224 × 224 pixels in size. The deconvolutional part is a mirrored version of the convolutional part and contains 13 layers on both the convolutional and deconvolutional side. The convolutional part is converged into two fully connected layers augmented at the end to impose class-specific projections. It is trained using a two-stage training procedure in which the first step involves training with easy examples. The second stage involves fine-tuning of the network learned in first stage with more challenging images. Our light-weight network consists of seven convolutional layers and three pooling layers towards the convolutional side. The deconvolutional network is the mirrored version of the convolutional network. The major modification in architecture [16] is the removal of some intermediate layers, including fully connected layers, which improves the computational complexity of the network. Both the architectures, DeconvNet and light-weight, are called encoder–decoder-based architectures, in which the convolutional part downsamples and the deconvolutional part upsamples the feature maps. Such architectures allow the use of max-pooling indices during upsampling which helps to obtain better segmentation maps with preserved global context information. However, the use of max-pooling indices slightly increases the computational cost. The original DeconvNet architecture and proposed light-weight network architectures are shown in Figure 1. The details, including each layer’s output and the kernel size of our light-weight network architecture, are shown in Table 1.

In the following two sections, we do a comparative study of the original and proposed network in terms of computational complexity and latency.

#### 3.1.1. Computational Complexity

For any CNN, the total computational complexity of the convolutional layer can be expressed as follows [39]:(3)$$O\left(\sum_{l=1}^{d}n_{l-1}s_{l}^{2}n_{l}m_{l}^{2}\right).$$

In Equation (Equation 3), *l* represents the corresponding layer; $n_{l-1}$ represents the number of filters in the (l−1)th layer; $s_{l}$ represents the spatial size (length) of filter in the *l*th layer; and $m_{l}$ is the spatial size of the output feature map. DeconvNet consists of 13 convolutional layers and 13 deconvolutional layers, whereas the proposed light-weight network consists of seven convolutional and seven deconvolutional layers. Incorporating the fact that the convolution and deconvolution operations are the same in terms of computation, the overall computational complexity for both networks is shown in Table 2. The proposed light-weight network has a complexity 1.56 times lower compared to that of the original network. This reduction in complexity is in favor of the low latency requirement of autonomous driving.

#### 3.1.2. Frame Rate

The scene segmentation algorithms for autonomous driving require a frame rate as high as possible. In this work, we aimed to find a network architecture that provides a better frame rate without compromising the accuracy. We performed this test on a Nvidia Quadro GP100 GPU with 16G memory. In this setup, while maintaining the comparable accuracy, our proposed light-weight network has a frame rate of 21 Frames Per Second (fps), which is better than that of the original network (17.7 fps).

### 3.2. Training

The second part of this work is about actual learning and fine-tuning the network with synthetic and real-world datasets. We fine-tuned our proposed light-weight network with synthetic datasets as well as with a real-world dataset and report the result. Here, we explore the advantages and disadvantages of using a synthetic dataset. We used the synthetic dataset as the intermediate domain and the real-world dataset as the final domain. In the first training method, we performed transfer learning using only the real-world data and observed the results. In the second training technique, we trained the light-weight network using the synthetic dataset as an intermediate domain. In this work, we are interested in seeing the effectiveness of our segmentation results in a real-world scenario by fine-tuning the light-weight network trained with synthetic dataset. To do so, we fine-tuned the original model with the synthetic dataset as a first step, and transferred this knowledge for the real-world dataset as a final step. As we are interested in the off-road autonomous driving scenario, we focused on how the transfer learning works in order to segment the real-world dataset with and without using synthetic dataset.

In this work, we used the softmax loss as an optimization function available in Caffe framework [40]. This loss function is basically a multinomial logistic loss that uses softmax of the output in the final layer of the network. The softmax function is the most common function used in the output of CNNs for classification. It is used as a layer in CNN architecture that takes an *N*-dimensional feature vector and produces the probabilistic values as output in the range (0, 1). Considering $\left[x_{1},x_{2},x_{3},\ldots ,x_{N}\right]$ as the input to the softmax layer and $\left[o_{1},o_{2},o_{3},\ldots ,o_{N}\right]$ as its output, the input–output mapping occurs as in Equation (Equation 4).
(4)$$o_{i}=\frac{e^{x_{i}}}{\sum_{y=1}^{N}e^{x_{y}}}\;\forall i\in 1\ldots N.$$

Therefore, in the classification or segmentation of input images, the softmax layer produces the probabilistic values for all possible classes. On the basis of these probabilities, any test data (or pixel in the case of segmentation) is assigned to the class with the maximum probabilistic value. Consider $\left(x_{1},y_{1}\right),\left(x_{2},y_{2}\right),\ldots \ldots ,\left(x_{n},y_{n}\right)$ to be any *n* number of training points, where *x* denotes the training data and *y* denotes the corresponding label. In Caffe [40], the softmax loss is defined in a composite form by applying multinomial logistic loss to the softmax layer’s output. In Equation (Equation 5), the softmax loss is defined as a cost function to be optimized.
(5)$$J\left(\theta \right)=-\frac{1}{n}\left[\sum_{i=1}^{n}\sum_{j=1}^{c}1\;\left\{y_{i}=j\right\}\;log\frac{e^{\theta_{j}^{T}x_{i}}}{\sum_{k=1}^{c}e^{\theta_{k}^{T}x_{i}}}\right],$$
where *c* represents the total number of classes. The parameter $\theta^{T}$ represents the transpose of the weight matrix of the network at that instant in time. With this loss function, the training was performed using the stochastic gradient descent method with a learning rate of 0.001, a momentum of 0.9, and a weight decay of 0.0005 in Nvidia Quadro GP100 GPU with 16G memory.

## 4. Experiments and Results

### 4.1. Dataset Description

We performed experiments with four different datasets in which three are simulated datasets and one is a real-world off-road dataset. The simulated datasets range from simple two-class datasets to more complex four-class datasets. These datasets were generated considering different real-world aspects such as surface reflectivity of tree trunks or the ground, the shadowing effect, time of the day, etc. The real-world dataset is the off-road autonomous vehicle dataset called Freiburg Forest dataset [41]. In the section below, we describe each of them briefly.

#### 4.1.1. The Synthetic Dataset

Three sets of synthetic data were used which were generated using a specially designed simulator enabled by the MSU Autonomous Vehicle Simulator (MAVS) [42,43]. This simulator is a physics-based sensor simulator for ground vehicle robotics that includes high-fidelity simulations of LiDAR, cameras, and several other sensors. In this work, these datasets are considered to assess the performance of segmentation, transferring the knowledge from the pre-trained convolutional network to the simulated dataset. In addition, we assess the segmentation performance by transferring the knowledge from the synthetic dataset to a real-world off-road driving scenario. As the off-road vehicle domain has very little data to use for training and it is a domain requiring the highest possible level of accuracy, a larger volume of annotated datasets are required. In order to fulfill this requirement, the use of a synthetic dataset can be a help.

##### The Two-Class Synthetic Dataset

This dataset is the simplest synthetic dataset containing two classes: Ground and Tree. This dataset does not strongly incorporate the characteristics of real-world scenes such as time of the day, shadowing, reflectivity, etc. However, it considers the properties of tree trunks, leaves, and the ground mostly in terms of color and structure. The dataset consists of 5674 images of size 640×480 pixels. We separated 80 percent into the training set and 20 percent into the validation set with no overlap. Some samples of this dataset are shown in Figure 2.

##### The Four-Class High-Definition Dataset

This dataset is more complex than the previous two-class dataset. It considers a more complex environment including vegetative structure as well as more realistic forest scenes. The increased complexity of this dataset is mostly due to fine vegetative structures sparsely distributed on the ground. Additionally, we consider the flowering as well as non-flowering vegetation and trees, making this dataset both realistic and complex at the same time. It simulates sky, trees, vegetation, and the ground as four different classes. In total, we have 1700 high-definition images of size 1620×1080 pixels; we separated 80 percent into the training set and 20 percent into the validation set with no overlap. Some typical images from this synthetic dataset are shown in Figure 3.

##### The Four-Class Random Synthetic Dataset

Compared to the other two synthetic datasets, this dataset is more natural as more real-world variations are considered. This also includes sky, trees, vegetation, and the ground as the classes for off-road driving scene. We use total 10,726 images of 224×224 pixels. In the MAVS simulator, the use of randomized scenes with physics-based simulation of cameras and environments allows for the use of a wide variety of training data. MAVS considers features such as different terrain structure, different time of the day, and haziness of the atmosphere quantified by turbidity [43]. As mentioned in [43], five different times of the day and five different turbidity values are considered, producing 25 unique lighting scenarios in the images. On the other hand, the random dataset includes images from three different environments: an American Southeast forest ecosystem, an American Southeast meadow ecosystem, and an American Southwest desert ecosystem. Because of this set up, this dataset has much more variance than the previous two synthetic datasets. Some sample images from this dataset are shown in Figure 4.

#### 4.1.2. The Real-World Dataset

We use Freiburg Forest dataset [41] as real-world dataset. These were collected at 20 Hz with a resolution of 1024×768 pixels on three different days to acquire the variability in data caused by lighting conditions. However, in our experiments, we pre-processed the dataset as per our requirement. Before feeding them into our proposed light-weight network, the images were cropped into 224×224 size as a pre-processing step to make them compatible with the input layer as well as to acquire simple data augmentation. In [40], cropping can be performed randomly to extract an image patch of a desired dimension. The images in the dataset are in different formats such as RGB, NIR, depth images. For this work, we use the RGB image format only. The dataset includes six different classes: Obstacle, Trail/Road, Sky, Grass, Tree, and Vegetation. While experimenting, we considered the tree and vegetation as a single class as suggested in [41]. Therefore, in terms of training, it is only a five-class dataset. Some sample images from this dataset pool are shown in Figure 5.

### 4.2. Segmentation of the Real-World Dataset with Transfer Learning

In this experiment, we train our light-weight network with the pre-trained weights from DeconvNet architecture. The DeconvNet architecture was originally trained with the Pascal VOC dataset (as a benchmark dataset for segmentation). To transfer the knowledge from this architecture, we initialize our proposed network with the pre-trained weights from DeconvNet corresponding to the existing layers in the light-weight network while ignoring the weights of the remaining layers. We apply fine-tuning by learning up to two layers completely from scratch towards the deconvolutional side of our light-weight network.

The proposed algorithm achieved 93.1 percent overall accuracy with only the outermost layer learning from scratch and 94.43 percent accuracy with the two outermost layers learning from scratch with a learning rate of 0.01. All the other layers are slowly modified with a learning rate of 0.001. This way of fine-tuning typically means adopting the DeconvNet to the new domain, where the general properties are slowly modified/learned and specific properties are quickly modified/learned. The concern about which layers are to be learned from scratch is an open-ended question and is mostly the function of diversity between the source and target domain. The results produced by the model with the best accuracy (the one that is trained with the two outermost layers learned from scratch) are shown in Figure 6.

### 4.3. Utilizing the Synthetic Dataset

In this experiment, we use TL approach somewhat differently. This training approach is based on training the network multiple times with multiple domains in order to slowly learn the target domain. We use three different synthetic datasets as the intermediate domain and observe the performance of fine-tuning for the real-world dataset. The obtained overall accuracy of segmentation for all three sets of synthetic datasets are shown in Table 3. The accuracy of the four-class high-definition dataset is lower compared to the other two synthetic datasets. As specified in Section 4.1.1, this dataset has the fine vegetative structures sparsely distributed on the ground which makes them difficult to detect. In addition, the vegetation and the trees are with and without flowers, which makes this dataset realistic and complex at the same time. This complexity inherent to the four-class synthetic dataset resulted into the lower accuracy.

#### 4.3.1. The Two-Class Synthetic Dataset

In this experiment, we first trained our proposed light-weight network with the pre-trained DeconvNet weights using the two-class synthetic dataset. As we specified in the earlier section, this dataset contains trees and ground as two classes and is a simple dataset. This dataset just considers the autonomous driving scenario in terms of color. The tree class is represented with a gray and green color, and the ground with a yellowish color, as shown in Figure 2. Structurally, the trees have minor variations and the ground is uniform. After training and testing with our proposed light-weight network, we obtained 99.15 percent overall pixel-wise accuracy with this synthetic dataset. We used the learning rate of 0.01 for the two outermost layers and 0.001 for all the other layers. We show some segmented results of this synthetic dataset in Figure 7.

The model trained with this two-class synthetic dataset is again fine-tuned with the real-world Freiburg dataset. This time, only the outermost layer of the light-weight network was learned from scratch with a learning rate of 0.01 and all the other layers with 0.0001. As shown in Table 4, we obtained 94.06 percent overall accuracy, which is somewhat below the accuracy given by the previous experiment.

#### 4.3.2. The Four-Class High-Definition Dataset

In this experiment, we trained our light-weight network with the pre-trained DeconvNet weights using the four-class synthetic dataset. This dataset is more complex than the two-class synthetic dataset in terms of the number of classes and their structure. The four classes in this dataset include ground, vegetation, tree, and sky. The vegetation includes smaller grass and/or bush like structures and contains variations such as flowers or no flower within it. After training and testing our proposed light-weight network with this dataset using the same learning rate setup as in the two-class dataset, we obtained 75.71 percent overall test accuracy. Some results of segmentation of the synthetic dataset are shown in Figure 8.

As in the previous experiment, the model trained with the four-class high-definition dataset is fine-tuned using the Freiburg Forest dataset. Only the outermost layer of the light-weight network was learned from scratch with a learning rate of 0.01 and all the other layers with 0.0001. We obtained the improved segmentation performance when compared with the results that did not use the synthetic dataset as well as with that of the two-class synthetic dataset. This improvement is obvious as the four-class high-definition dataset considers more realistic properties of the real-world environment in terms of number of classes and intra-class variability.

#### 4.3.3. The Four-class random synthetic dataset

In this experiment, we trained our proposed light-weight network with the pre-trained DeconvNet weights using the four-class random synthetic dataset. We used the same learning rates as in the experiment with the two-class synthetic dataset. As specified earlier, this dataset is complex in the sense that it considers different factors to make it more realistic. Some factors considered are different time of the day, different terrain surface, varying tree structure, etc. As in the dataset used in previous experiment, it also contains four classes including ground, vegetation, tree, and sky. After training and testing our proposed light-weight network with this dataset, we obtained 91 percent overall accuracy. Some of the results of the segmentation of the synthetic dataset are shown in Figure 9.

Again, as with the four-class high-definition dataset, the model trained with the four-class random synthetic dataset is fine-tuned using the real-world Freiburg dataset. In this part, only the outermost layer of the light-weight network was learned from scratch with a learning rate of 0.01 and all of the other layers with 0.0001. As shown in Table 4, the performance of the light-weight network for transfer learning with this dataset decreased somewhat compared to the previous three experiments. As stated above, consideration of the real-world properties for the forest environment is increased in this dataset. However, the reduction in the overall accuracy could be due to increased variation among the dataset that caused the network to learn the features that are less correlated to the target domain. This phenomenon is sometimes called negative transfer. In Figure 10, we show the comparative results for all the experiments.

## 5. Result Analysis and Discussion

Table 4 shows the comparative results of the proposed method including the baseline [16] method in terms of overall accuracy. We can analyze these results in terms of two aspects: network and TL method. Our proposed light-weight network gives much better results compared to the DeconvNet for all the four experiments. Most surprisingly, the results obtained are much better after stripping down the network to a half of its original size. This result favors the requirement of autonomous driving which needs higher accuracy with reduced latency. On the other hand, if we analyze the table in terms of TL method, we can see mixed results. For the TL with DeconvNet, the use of the synthetic dataset as intermediate domain led to a negative performance. Whereas, with our proposed light-weight network, we achieved an increased performance after using the four-class high-definition datasets compared to that which did not use the synthetic dataset. For both the datasets, the two-class synthetic and the four-class random synthetic, the performance decreased slightly. The two-class synthetic dataset is a simpler dataset which does not take into account the real-world environmental effects in terms of both the number of classes and their properties. This dataset just increased the volume with no helpful information learned before moving into the target domain causing negative transfer performance. On the other hand, the random dataset includes images from different environments. It includes the data from three different environments: an American Southeast forest ecosystem, an American Southeast meadow ecosystem, and an American Southwest desert ecosystem with their various lighting conditions. These different environments caused a high level of randomness and a lower correlation to the target domain; this dataset also added no helpful knowledge while doing the transfer learning. However, it also caused the negative transfer. The four-class high-definition dataset gave the positive TL performance with the accuracy of 94.59% on the Freiburg test set. Different from the two other datasets, this dataset has higher correlation with the target domain. Additionally, the huge randomness caused by the various ecosystems in the four-class random dataset is not available in the four-class high-definition dataset. The forest and ground structure have comparatively more similarity with that of the target domain which causes the improved performance while training with the Freiburg dataset.

We show the confusion matrices for each experiment performed with the proposed light-weight network in Table 5. Each entry is the percentage measurement of either the correctly or falsely classified number of pixels in all test images. We can see the obstacle class having the lowest accuracy and the sky class having the highest accuracy in each TL experiment. The cause for the low accuracy regarding obstacles is that the pixels belonging to this class are very limited in the training datasets compared to the other classes. In addition, the obstacle in the training images have less structural uniformity. This results in the network learning less about the obstacle class causing a biased prediction in favor of classes having a higher number of pixels.

## 6. Conclusions and Future Work

In this paper, we explored the transfer learning from the perspective of network size and training techniques with and without the use of synthetic data. We conclude that it is important to find out the size of the network that performs best for the target domain rather than using the original architecture as a whole. In doing so, we proposed a new light-weight network; a network well suited for use in autonomous driving applications due to its low latency, which is initialized with the pre-trained DeconvNet weights from the corresponding layers. Furthermore, we explored the effects of using different synthetic datasets as the intermediate domain. As TL techniques are used for these domains where training datasets are insufficiently available, generating and using synthetic datasets is a good approach, which can help boost performance. While doing so, considering the target domain characteristics as much as possible when generating the synthetic dataset will increase the TL performance. We also conclude that an oversimple and/or too random dataset, as was the case for the two-class synthetic and the four-class random synthetic dataset herein, can cause negative transfer.

The intermediate layers and their weights of DeconvNet are absent in the proposed light-weight network. In order to understand the relationship among the layers and correspondence between layers from source to target network, a detailed theoretical study is needed focusing the semantic meaning, i.e., mapping between features across layers of the target and source domain. While there exists some work to understand what the features means in different layers—e.g., initial layers extract lower level features—for classification task, there is no such study for encoder–decoder architecture targeted for segmentation task. In the future, we plan to study the detailed theoretical underlying regarding those aspects for encoder–decoder-based networks. This would also shed light on how the proposed way of transfer learning leads to better adaptability and performance. Furthermore, we plan to incorporate our road segmentation model into the real off-road autonomous vehicle and study the creation of occupancy grid with the segmentation results to support decisions of path planning.

## Figures and Tables

**Figure 1 sensors-19-02577-f001:**
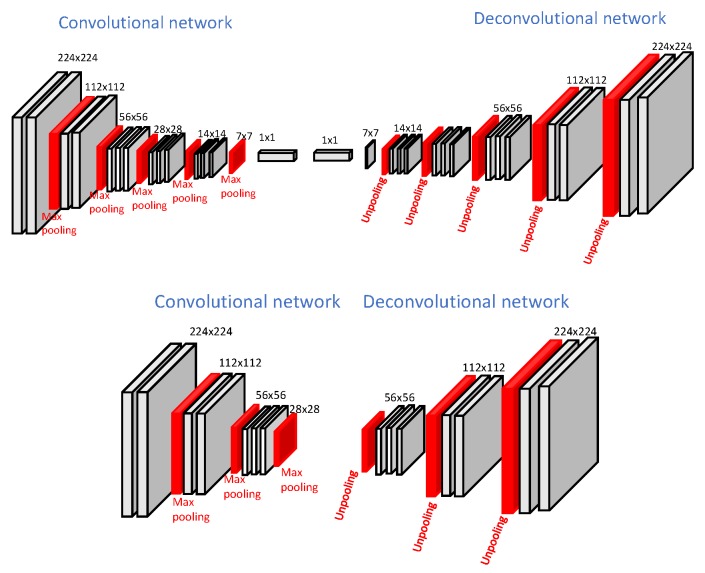
**Top**: Original DeconvNet architecture, **Bottom**: Proposed light-weight network architecture.

**Figure 2 sensors-19-02577-f002:**

Sample images from two-class synthetic dataset. Best viewed in color.

**Figure 3 sensors-19-02577-f003:**
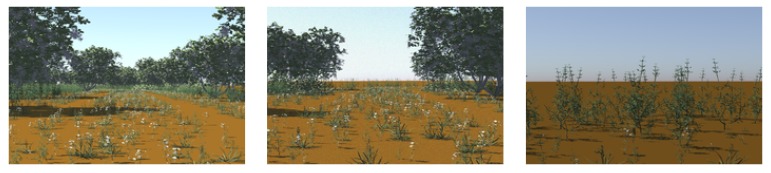
Three sample images from four-class high-definition dataset. best viewed in color.

**Figure 4 sensors-19-02577-f004:**
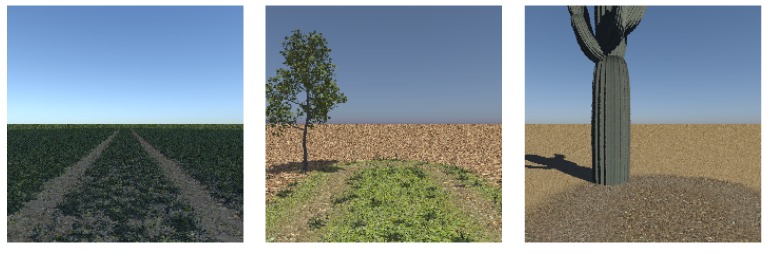
Sample images from four-class random synthetic dataset. Best viewed in color.

**Figure 5 sensors-19-02577-f005:**
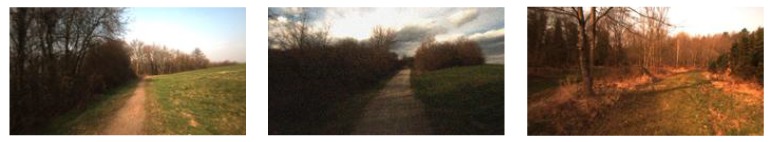
Sample images from Freiburg Forest dataset. Best viewed in color.

**Figure 6 sensors-19-02577-f006:**
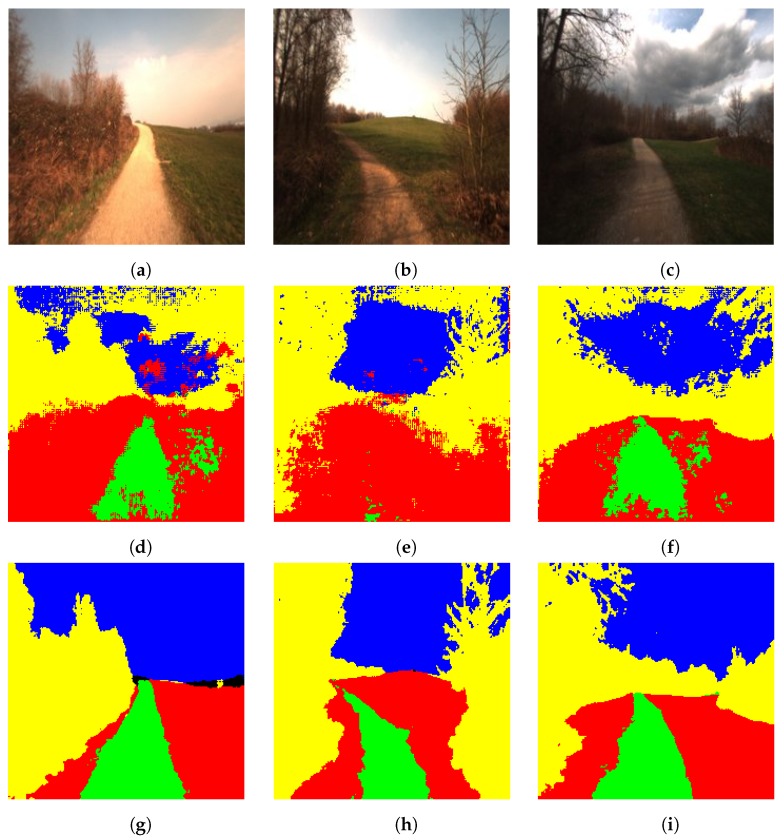
Segmentation of the Freiburg Forest dataset (**a**–**c**): test images, (**d**–**f**): corresponding segmented images using DeconvNet, (**g**–**i**): corresponding segmented images using the proposed light-weight network. Note that the color code for classes is: yellow: tree, green: road, blue: sky, red: ground, black: obstacle. Best viewed in color.

**Figure 7 sensors-19-02577-f007:**
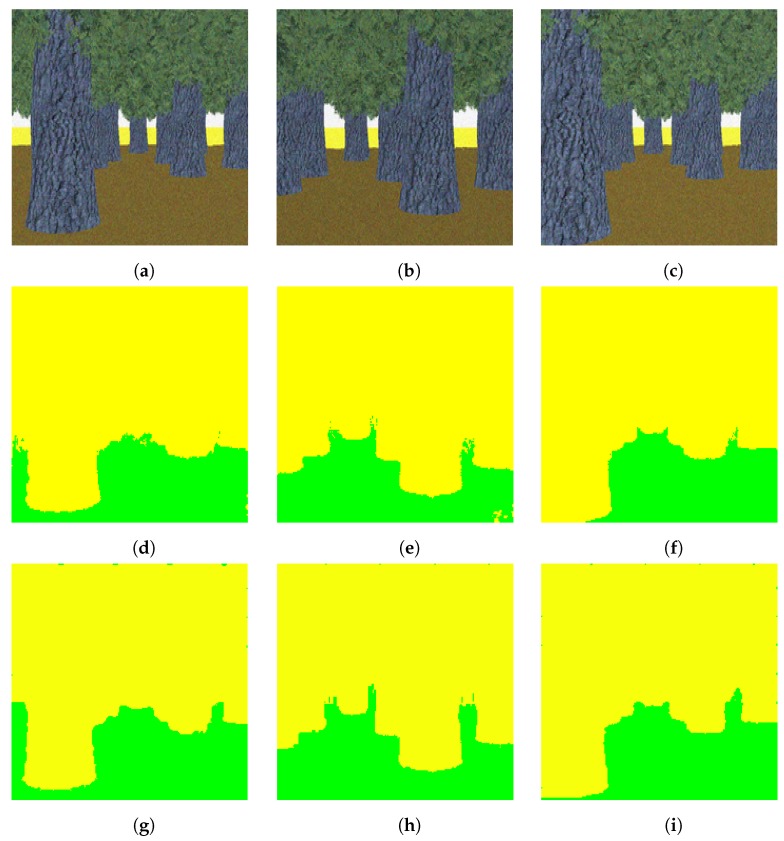
Segmentation of the two-class synthetic dataset (**a**–**c**): test images, (**d**–**f**): corresponding segmented images using DeconvNet, (**g**–**i**): corresponding segmented images using the proposed light-weight network. Note that the color code for classes is: yellow: tree, green: ground. Best viewed in color.

**Figure 8 sensors-19-02577-f008:**
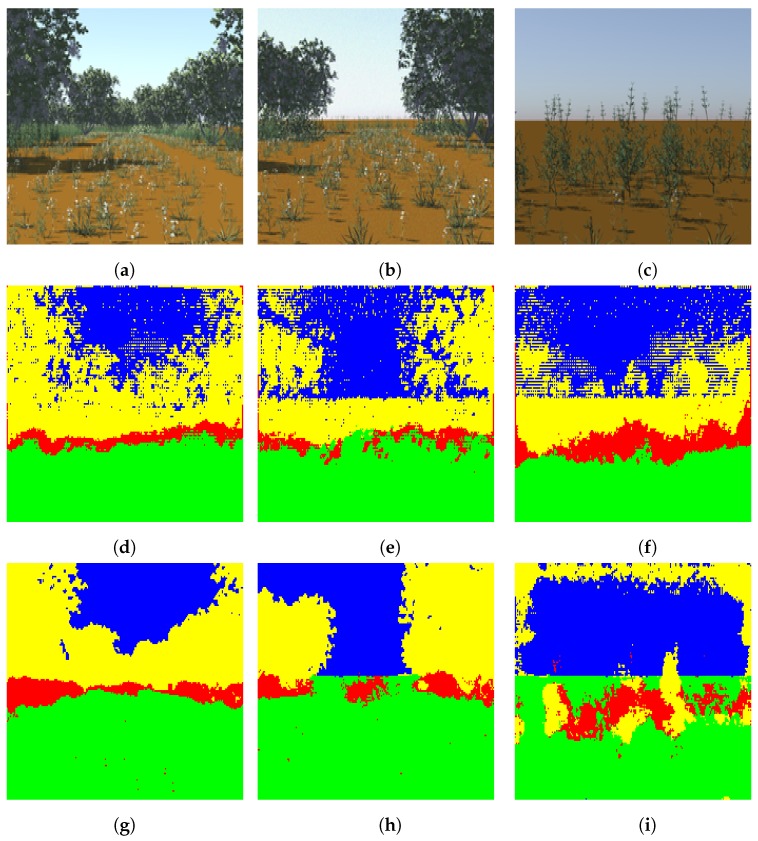
Segmentation of the four-class high-definition dataset (**a**–**c**): test images, (**d**–**f**): segmented images using DeconvNet, (**g**–**i**): segmented images using proposed light-weight network. Note that the color code for classes is: green: ground, red: vegetation, yellow: tree, blue: sky. Best viewed in color.

**Figure 9 sensors-19-02577-f009:**
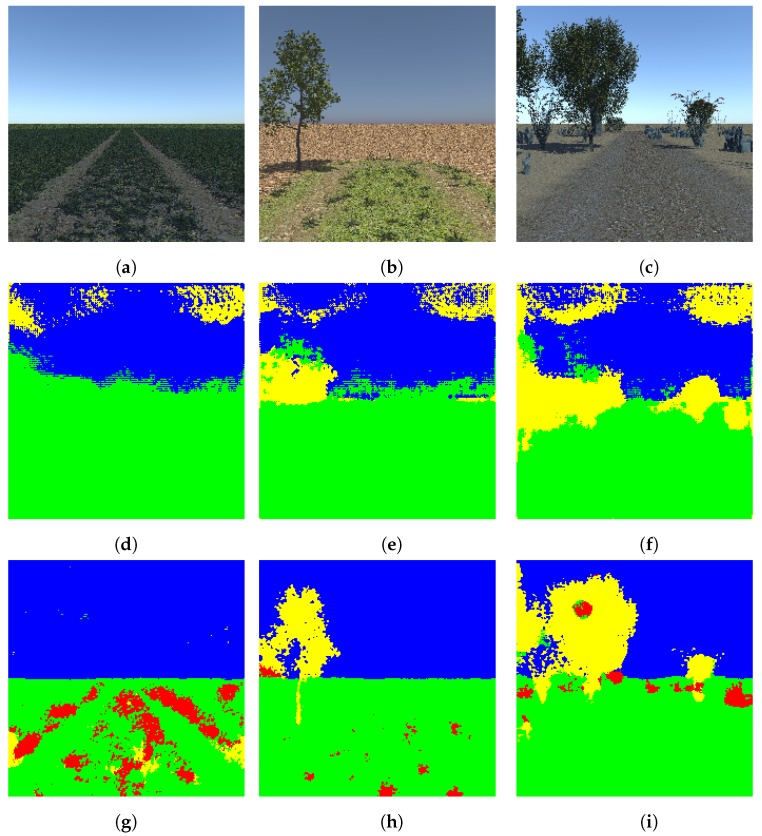
Segmentation of the four-class random dataset (**a**–**c**): test images, (**d**–**f**): segmented images using DeconvNet, (**g**–**i**): segmented images using proposed light-weight network. Note that the color code for classes is: green: ground, red: vegetation, yellow: tree, blue: sky. Best viewed in color.

**Figure 10 sensors-19-02577-f010:**
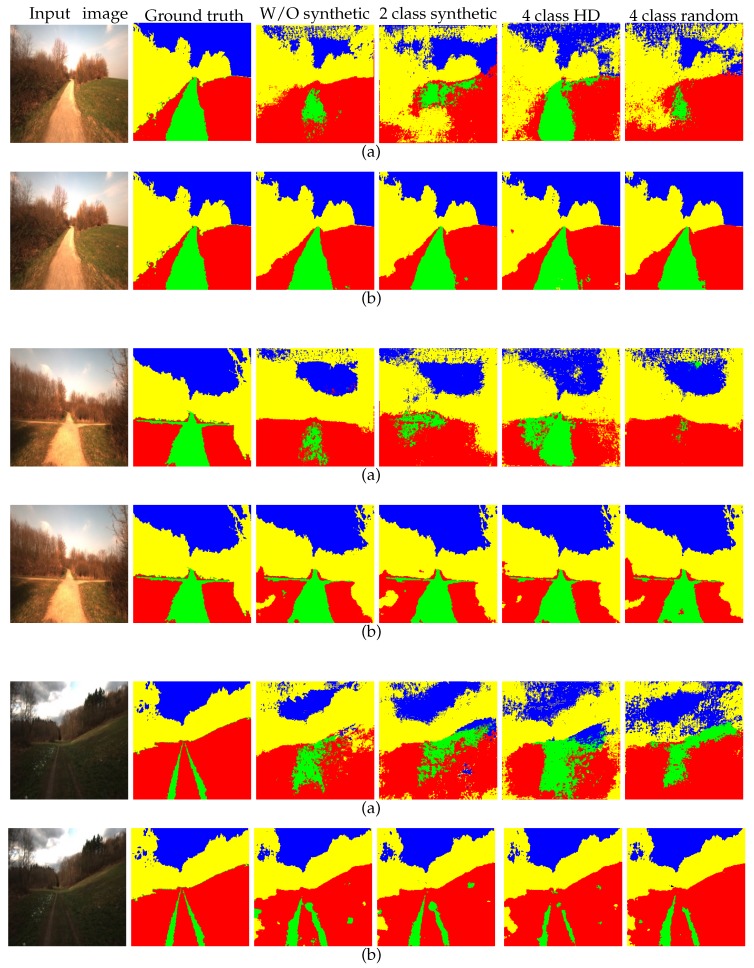
Examples to show that the light-weight network produces better results than DeconvNet for each of the experiments. Note that each pair of rows (**a**,**b**) represents the results produced by DeconvNet and the proposed light-weight network, respectively. Best viewed in color.

**Table 1 sensors-19-02577-t001:** Detailed structure of proposed light-weight network architecture. Note that *C* is the number of classes.

Layer’s Name	Kernel Size	Stride	Pad	Output Size
input	-	-	-	224×224×3
conv1-1	3×3	1	1	224×224×64
conv1-2	3×3	1	1	224×224×64
pool1	2×2	2	0	112×112×64
conv2-1	3×3	1	1	112×112×128
conv2-2	3×3	1	1	112×112×128
pool2	2×2	2	0	56×56×128
conv3-1	3×3	1	1	56×56×256
conv3-2	3×3	1	1	56×56×256
conv3-3	3×3	1	1	56×56×256
pool3	2×2	2	0	28×28×256
unpool3	2×2	2	0	56×56×256
deconv3-1	3×3	1	1	56×56×256
deconv3-2	3×3	1	1	56×56×256
deconv3-3	3×3	1	1	56×56×128
unpool2	2×2	2	0	112×112×128
deconv2-1	3×3	1	1	112×112×128
deconv2-2	3×3	1	1	112×112×64
unpool1	2×2	2	0	224×224×64
deconv1-1	3×3	1	1	224×224×64
deconv1-2	3×3	1	1	224×224×64
output	1×1	1	1	224×224×C

**Table 2 sensors-19-02577-t002:** Complexity comparison of the two networks.

Network	Complexity	Ratio
DeconvNet	$O\;(2.914\times 10^{10})$	O (1.56)
Light-weight	$O\;(1.867\times 10^{10})$	

**Table 3 sensors-19-02577-t003:** Segmentation accuracy on the synthetic dataset. TL—Transfer Learning.

Data	Method	DeconvNet	Light-Weight
Synthetic	TL on two-class	97.62 (%)	99.15 (%)
	TL on four-class high-definition	65.61 (%)	75.71 (%)
	TL on four-class random	73.23 (%)	91.00 (%)

**Table 4 sensors-19-02577-t004:** Quantitative results produced by DeconvNet and the proposed network for various TL experiments. Shading indicates the improvement of one method over another.

Data	Method	DeconvNet	Light-Weight
Freiburg	W/O synthetic data	73.65(%)	94.43(%)
	After using two-class synthetic	66.62(%)	94.06(%)
	After using four-class high-definition	68.7(%)	94.59(%)
	After using four-class random synthetic	68.14(%)	93.89(%)

**Table 5 sensors-19-02577-t005:**
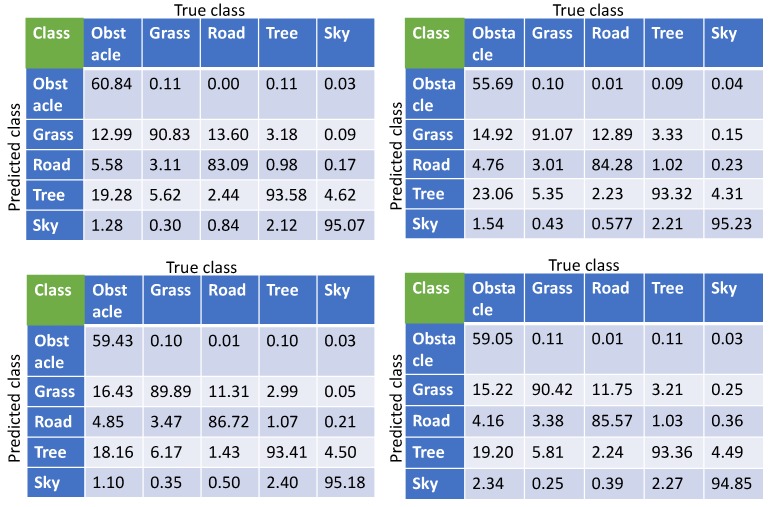
Confusion matrices of the test results produced by the proposed network for different TL experiments on the Freiburg Forest dataset. Note that each entry is an overall percentage. (Top left: without using synthetic dataset, top right: using two-class synthetic dataset, bottom left: using four-class high-definition synthetic dataset, bottom right: using four-class random synthetic dataset).

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
