# Peer review of "Semantic Segmentation with Transfer Learning for Off-Road Autonomous Driving"

_sensors, 2019, doi:10.3390/s19112577_

Round 1

Reviewer 1 Report

The paper has two goals in semantic segmentation: 1) Show that a stripped-down variant of DeconvNet runs faster with good performance on simpler problems. The light-weight variant simply removes several narrow layers near bottleneck of the network. 2) Show if intermediate TL by fine-tuning of network with synthetic datasets improves performance. Several options are compared (plain or realistic rendering).

The paper is clearly written and addresses a relevant problem of speeding up algorithms for real-time AD application.

1) Authors claim 18.55x computational complexity gain, which seems rather high given that only the narrow half of network is removed (Fig.1) - check the calculation. 

The reported framerate improvement is however only 1.2x (Sec.3.1.2), which is in striking contrast with above theoretical estimate, but also does not constitute a significant difference to the AD application.

2) Proposed synthetic TL scheme fails to have a significant impact on the resulting accuracy (Tab.4), and the two datasets with overly simplified renderings even give negative performance gain (not to much surprise).

The presented experiments suggest that the methods segments reasonably well on the real dataset, but neither quantitative nor qualitative comparison of the proposed lightweight network with the baseline [16] or other state-of-the-art (eg. Deeplab) is given.

Authors must extend their experiments to allow assessing the contribution of the paper, in particular all Tabs.3,4 and Figs.6-10 should show baseline.

Minor:

eq.1 use \mid in conditional prob.

eq.3 i=1  ->  l=1 

Fig.6-10: unify label colors across datasets

Reviewer 2 Report

This paper conducts a transfer learning approach to use an adapted DeconvNet network to be used in a segmentation problem regarding off-road driving. Additionally, the authors explore the addition of synthetic data in regards to improvement in accuracy. However, my main concern is that novelty of the paper is not sufficient for publication. All used techniques and datasets have been used in previous literature, and demonstrations about synthetic data, as well, as transfer learning already made.

Additionally, here are some additional notes:

line / note

275: cropped to 224x224 (how is the process performed?)

284: can you explain how the pre-trained weights could be loaded in your network if other intermediate layers were eliminated? Please add references or ellaborate.

Figs. 8-9: colors are inconsistent, please redo. Also, aren't vegetation and trees under the same class as stated in the text?

Table 4: Please explain why results are worse with the "4-class random" intermediate dataset. One sentence in the conclusions seems insufficient.

Furthermore, please check the notes for detected typos and other issues found on the paper. (Legend: typos in red, concerns in orange, personal highlights in yellow).

Reviewer 3 Report

1.        Authors presented the transfer learning approach to identify the classes form the scene by additionally training the model with synthetic datasets and performs the upsampling with deconvolution. For learning the scene from the images, authors choose the existing deep learning model known as DeconvNet. In the script, authors claim that their network is lightweight in terms of layers as compared with the DeconvNet. However, it may not be expected to have smaller data for the training in order to correctly classify the segments of a scene, how do you think this issue or how to cope with this problematic issue?  If this is not a problematic issue, please justify its feasibility or the rationality.

2.        The transfer learning research area focused in the papers is a very interesting and overall the manuscript is written in a good way. It would be suggested to add and update the followings.

-        Authors should discuss that why  DeconvNet is used for their application instead of Alexnet, VGGNet etc.

-       What are the  pros and cons to using autoencoder/decoder based  CNN model in semantic segmentation application in TL

-       What are the ensemble techniques, are the ensemble based technique improve the TL semantic segmentation model performance in terms of processing and accuracy. Please discuss it in the manuscript

-       Add description and motivation of using synthetic Pascal visual object classes dataset in the introduction part and experimental part of the manuscript

-   As authors claim that decreasing the network size improve the performance of the model in TL. However in one hand author decrease the network size and on the other hand, they feed more training data to the model in order to improve the accuracy of the model. The motivation and rationale should be discussed in the introduction, discussion and conclusion part of the manuscript. This addition will increase the interest of authors in the manuscript.

-        In line, no 288 authors mention several learning rates. It would be plus to add details of learning rates used in the experiment.

-       In table 3 the accuracy of the second dataset is lower than the others, discuss the factors  in details that contribute to the lower accuracy for this case

-       The detail of the Parameters and attributes of the datasets used in the experiment should be discussed in details. For this, a single combined table can be added which shows the detail and number of attributes of the datasets used in the experiment.

-       I would like to ask authors to fix the number and labels of figures and refer it correctly in the manuscript.  As at line number 316 may be wrong figure number or label is referred.

Round 2

Reviewer 1 Report

Authors have fixed the minor comments and improved presentation.

I welcome the expanded experimental comparison and it formally looks OK, but I am not really convinced where the performance gain over the added baseline comes from.
The expected result for a light-weight version is that the performance will be slightly worse than the full-blown original, or about the same at best, yet the full version is actually shown to be 10-30% worse. This could potentially be due to the lack of training data needed for a more complex network, but at least for the purely synthetic results, where training data is virtually unlimited, this should not be the case.

That raises doubts if the full DeconvNet used for comparison was fine-tuned similar to the proposed (which is also not explicitly written in the paper). 

For the final version I therefore ask authors to verify the baseline results and comment on the surprisingly bad results of the baseline, preferably based on additional experiment involving larger training set similar to original DeconvNet paper.

I agree with the criticism of the second reviewer regarding the paper novelty, which should be compensated by sound experimentation and result analysis, as suggested above.

Author Response

See the attached document.

Reviewer 2 Report

I understand the authors have made a good effort in improving the paper as per the suggestions received. However, I still have some concerns regarding the transfer of weights. I do understand that weights can be pre-loaded using whichever tool available (be it in Caffee or in Keras, "thawing" a frozen model, etc). My concern was "semantic" rather than technical. I would suggest the authors add a reference or two to encoder-decoder TL literature, so they can provide a citation/justification as to the fact that cropping the latest layers of the encoder, and the earliest layers of the decoder is feasible. It seems logical that it should be, but someone must have made a study justifying precisely this point. Nonetheless, the method is better explained now, its originality though is still not much. So I leave it to the editor to decide whether it is sufficient for publication on their journal.
